# Figure–Ground Segmentation and Biological Motion Perception in Peripheral Visual Field

**DOI:** 10.3390/brainsci13030380

**Published:** 2023-02-22

**Authors:** Ilze Ceple, Jurgis Skilters, Vsevolod Lyakhovetskii, Inga Jurcinska, Gunta Krumina

**Affiliations:** 1Department of Optometry and Vision Science, University of Latvia, LV-1586 Rīga, Latvia; 2Laboratory for Perceptual and Cognitive Systems, Faculty of Computing, University of Latvia, LV-1586 Rīga, Latvia

**Keywords:** biological motion, visual periphery, cortical magnification

## Abstract

Biological motion perception is a specific type of perceptual organization, during which a clear image of a moving human body is perceptually generated in virtue of certain core light dots representing the major joint movements. While the processes of biological motion perception have been studied extensively for almost a century, there is still a debate on whether biological motion task performance can be equally precise across all visual field or is central visual field specified for biological motion perception. The current study explores the processes of biological motion perception and figure–ground segmentation in the central and peripheral visual field, expanding the understanding of perceptual organization across different eccentricities. The method involved three different tasks of visual grouping: (1) a static visual grouping task, (2) a dynamic visual grouping task, and (3) a biological motion detection task. The stimuli in (1) and (2) were generated from 12–13 dots grouped by proximity and common fate, and, in (3), light dots representing human motion. All stimuli were embedded in static or dynamics visual noise and the threshold value for the number of noise dots in which the elements could still be grouped by proximity and/or common fate was determined. The results demonstrate that biological motion can be differentiated from the scrambled set of moving dots in a more intensive visual noise than static and dynamic visual grouping tasks. Furthermore, in all three visual tasks (static and dynamic grouping, and biological motion detection) the performance was significantly worse in the periphery than in the central visual field, and object magnification could not compensate for the reduced performance in any of the three grouping tasks. The preliminary results of nine participants indicate that (a) human motion perception involves specific perceptual processes, providing the high-accuracy perception of the human body and (b) the processes of figure–ground segmentation are governed by the bottom-up processes and the best performance can be achieved only when the object is demonstrated in the central visual field.

## 1. Introduction

Perceptual organization refers to the selection, grouping, and extraction of the available visual information and forming a percept of a single object with definite features. While these features are determined by each part of the object, the whole object might have different overall characteristics and qualities when compared to its parts taken individually [1]. Starting from Wertheimer, the core processes and principles of perceptual organization have been studied extensively, i.e., visual grouping, shape assignment, and object formation according to features of separate elements, such as proximity, motion direction, symmetry, parallelism, good continuation, closure similarity in color, orientation, or size to mention just a few [1,2,3,4]. Furthermore, perceptual organization involves the two main processes of information analysis: perceptual grouping and figure–ground segmentation [1]. The process of perceptual grouping determines which items in visual input will perceptually bind together into whole objects, while figure–ground segmentation determines the shape and spatial characteristics (e.g., orientation) of the object—where the whole object in relation to other objects is located. Of course, although visual grouping and shape assignment are at least to some extent hierarchically ordered and interrelated processes, they are most likely to occur at different stages of perceptual processing. In some cases, perceptual grouping might be early, whereas in others—parallel with shape assignment or in other cases still—a late process that is co-determined by previous experience [3]. Independently of whether grouping is slow or fast or an early or late process, it is the most crucial part in the processes of perceptual organization, inducing the part-whole dependencies in visual field.

Biological motion perception is a specific task of perceptual organization, during which percept of a moving human body (or an animal) is formed based only on the information on the major joint movements [5]. The 12–13 dots representing human locomotion are sufficient for not only perceiving human motion but also determining the action category [6], object gender [7,8], object intentions [9], emotions [10], and even person recognition [11]. According to Chang and Troje, biological motion perception can be either based on the local cue analysis (the motion of separate joints) or on the global information analysis (the motion of the whole body and real-time changes in location of all joints in space) [12].

The performance of different visual perception tasks (including perceptual organization) is not equally accurate across all visual field: the ability to distinguish detailed objects is the highest when the object image is projected on fovea (central visual field) and the performance deteriorates towards the visual periphery (when the object image is projected on the peripheral retina) [13]. Different studies have demonstrated that decreased performance in the visual periphery is observed in tasks requiring high visual acuity [14], contrast sensitivity [13], stereo vision [15], color vision [16], motion perception [17], word recognition [18], face recognition [19], and other visual functions.

Decreased task performance towards the visual field periphery is attributed to structural differences in the central and peripheral retina, as well as differences in the neural level of visual information analysis from central and peripheral visual field. The number of primary cortex neurons representing the central visual field is significantly higher than neurons representing visual periphery [20,21]. By taking into account the cortical magnification factor, Johnston and Wright demonstrated that the reduced performance in motion detection in the near periphery can be equalized to the performance in central visual field if sufficient stimulus magnification is applied [22,23]. This is also true for visual acuity [22], vernier acuity [24], contrast sensitivity [25], and the performance of different visual search tasks in peripheral visual field [26]. However, there are some visual tasks (e.g., letter recognition) that cannot be performed equally well in the central and peripheral visual fields even when the stimulus size has been adjusted by cortical magnification factor [27].

Likewise, the performance of perceptual organization is not equally high across all visual field. Hess and Dakin demonstrated that when detecting similar orientation, the highest performance can be observed only in the central 10 degrees of the visual field, and the performance in further visual periphery cannot be equalized to the central visual field [28]. Similar results were obtained by Nugent et al., demonstrating that the performance of contour integration deteriorates towards visual periphery when stimulus is presented more than 10 degrees eccentrically [29]. On the other hand, by using more simplified stimuli of grouping objects into circles or ovals, Kuai and Yu demonstrated that the performance of perceptual organization is constant up to 35 degrees of visual eccentricity [30].

In a comprehensive study on perceptual organization across visual field, Tannazzo et al. determined the performance of perceptual grouping by different grouping principles—similar orientation, luminance, motion direction, and proximity. In order to determine whether stimulus magnification is able to compensate for the reduced performance in the peripheral visual field, the size of the objects presented in the visual periphery was proportionally increased. As expected, the results indicated a significantly lower performance in grouping tasks when the objects were seen in visual periphery. Furthermore, the decreased performance varied across different tasks (different grouping principles): the performance of grouping by motion and orientation decreased more rapidly than grouping by luminance or proximity. By increasing the stimulus size, the performance could be equalized to the performance in a central visual field of up to 40 degrees of visual eccentricity, indicating that further visual periphery is not well specialized in perceptual organization [31].

A similar study analyzing the perception of biological motion in the central and peripheral visual field was conducted by Ikeda et al., who applied visual noise and determined the number of noise dots in which biological motion could still be differentiated from its scrambled version. According to their results, the perception of biological motion in the peripheral visual field was described as “unscalably poor”, indicating that stimulus magnification cannot compensate for reduced performance in the visual periphery, and the central visual field is necessary for biological motion perception [32]. Since biological motion perception in visual noise is based on the global information analysis of whole-body motion (i.e., not on the local analysis of separate joint movement), Bertenhal and Pinto [33] and Thompson et al. [34] attempted to determine whether the difficulties in biological motion perception in peripheral visual field are related to global information analysis. Although the reaction time for task performance increased towards the visual periphery, the participants were still able to determine the direction of biological motion up to 10 degrees of visual eccentricity. By adding the visual noise, Thompson et al. [34] observed that the performance towards the visual periphery deteriorated more rapidly.

Gurnsey et al. examined whether biological motion perception in the peripheral visual field could be altered by task difficulty. The study analyzed the performance of biological motion detection and motion direction discrimination tasks in the central and peripheral visual field. The results demonstrated that stimulus magnification was able to compensate for the reduced performance in the peripheral visual field in both tasks of biological motion analysis [35]. The differences from the results obtained by Ikeda et al. [33] were again linked to the visual noise applied in their study [33] and figure–ground segmentation. Finally, Gurnsey et al. extended their previous studies and analyzed the minimal angle of deviation from straight movement at which biological motion direction could still be determined. The results of the study again demonstrated that stimulus magnification can compensate for reduced performance in the visual periphery and the observers were capable of determining even a ±1.5 degree deviation from straight movement of the biological motion stimulus up to 16 degrees of visual eccentricity [36].

The performance of perceptual organization in central and peripheral visual field seem to depend on particular visual grouping tasks, type of stimuli, and the provided instructions. The impaired perception of biological motion perception on the peripheral visual field demonstrated by Ikeda et al. [33] might also be related to the processes of figure–ground segmentation. Taking into account the factors and constraints mentioned below, the aim of the current study is to determine the performance of different figure–ground segmentation tasks in central and peripheral visual field. Therefore, exploring whether the highest accuracy in figure–ground segmentation is only possible when attending the stimuli directly.

## 2. Materials and Methods

### 2.1. Participants

Nine participants (8 female, 1 male; 22–29 years old) participated in the study. All participants had normal or corrected to normal vision (VA = 1.0 at near, decimal system) and reported no neurological or psychiatric disorders that might affect the results of the study in general. The participants were not informed about the aim and objectives of the study until after performing the task. The study was approved by the Ethics Committee of the Institute of Experimental and Clinical Medicine Participants, University of Latvia, No. 13/03/2020. All participants filled a consent form and were informed of the methodology applied to the current research. Participants were free to ask any questions or withdraw from the study at any time.

### 2.2. Stimuli

Three visual grouping tasks were designed to analyze figure–ground segmentation processes in the central and peripheral visual field: (1) a static visual grouping task, (2) a dynamic visual grouping task, and (3) a biological motion stimulus. The static visual grouping task was designed as 12 equal dots distributed in either three (Figure 1a) or four (Figure 1b) columns or 12 dots distributed randomly across the same area without being grouped according to the law of proximity (the scrambled version of the stimulus, Figure 1c). The size of each dot in the scrambled version was equal to the size of the dots that were distributed in columns, thus, the angular size of each dot and whole stimulus was kept constant. The participants were asked to determine whether there were 12 dots on the screen that were grouped in columns by pressing the keyboard button “Y” if they agreed that the alignment of dots was based on columns. If their answer was negative, they were asked to press the button “N”. In case of not being sure, participants were instructed to make a guess.

The dynamic visual grouping task was designed as 12 dots grouped according to similarity, proximity, and common fate; additionally, the scrambled version of the stimulus was also used. The dots of the grouped version of the stimulus (Figure 2a) were distributed in three columns and all 12 dots moved in the same direction: up, down, and to the left or right. The scrambled version of the stimulus (Figure 2b) was also designed to be 12 dots; however, in this case, the initial position and motion direction of each dot was chosen randomly. The size of each dot in the scrambled version of the stimulus and the area where the objects were distributed was equal to the dot and area size of the grouped version of the stimulus, thus the angular size of each dot and the whole stimulus size was kept constant. The duration of the stimulus demonstration and the motion speed of each dot was equal to the duration and the average motion speed of the biological motion stimuli described later (9.5 degrees per second). The participants were asked to determine whether there were 12 dots that could be grouped in columns and were moving in the same direction by pressing the button “Y” on the keyboard if the answer was positive and “N” if the answer was negative. If participants were not sure, they were instructed to make a guess.

Biological motion stimuli were generated based on the Action database developed by Vanrie and Verfaillie [37]. Motion objects were represented by 13 dots, corresponding to the head, shoulders, elbows, hands, hips, knees, and feet of a walking human body. The object moved in one of five directions (90 degrees to the right, 90 degrees to the left, 45 degrees to the right, 45 degrees to the left and straight ahead (0 degrees)). Participants were presented with either the point-light walker performing one full gait cycle (Figure 3a) or its scrambled version (Figure 3b) [34,38]. The scrambled version of the stimuli was generated from 13 dots of the same size, individual moving trajectory, and speed as in the separate dots constructing the biological motion stimuli previously described. The differences between the normal biological motion stimuli and the scrambled version were that the dots in the scrambled version were randomly distributed across the same area as in the biological motion stimuli, the motion of each dot was taken from the upside-down version of the biological motion and the initial position of each dot was chosen randomly—the motion of all dots was not coherent in respect to human body as in normal biological motion. The local motion information was preserved; however, it was impossible the group the dots into a percept of a walking human body.

The participants were presented with either the point-light walker or its scrambled version and were asked to determine whether the demonstrated object was a human body. If the demonstrated stimulus was recognized as a human body walking in one of the five directions (Figure 3a), participants were asked to press button “Y” on the keyboard. If the object could not be recognized as a human body, participants were asked to press “N” on the keyboard. In case of uncertainty, participants were instructed to make a guess.

### 2.3. Visual Noise

In order to analyze the processes of grouping in the central and peripheral visual field, all previously described stimuli were embedded in visual noise. Each point of the visual noise had similar perceptual qualities (size, color, and motion trajectory) as the points of the corresponding scrambled version of each stimulus; however, the noise dots were distributed in a larger area across the stimulus (Figure 4).

All three parts of the experiment (the static visual grouping task, dynamic visual grouping task, and the biological motion stimulus) were presented in a separate stimulus set—the different grouping tasks were not mixed together. Participants were instructed to determine whether the demonstrated object was a visual grouping stimulus or its scrambled version, and depending on the response accuracy, the amount of visual noise was altered. The threshold value for the number of noise dots in which the object could be distinguished from the visual noise was calculated based on the block-up-and-down, two-interval, forced-choice psychophysical procedure BUDTIF [39]. Each stimulus level (number of noise dots) was demonstrated four consecutive times. If the response was correct all four times, the number of noise dots in the next trial was increased. If the participant gave at least one wrong answer, the number of noise dots in the next trial was decreased. The initial number of noise dots was set to 0 (the stimuli were demonstrated without any visual noise). If the participant was able to differentiate the stimulus from its scrambled version, the number of noise dots was increased to 20. After two reversals the step size was reduced to 10, and after the following two reversals the step size was reduced to 5. The step size of 20 and 10 points was considered as a preparatory phase to reach a closer value to the threshold value (similar to the method described by Vleugels et al. [40]). The threshold value for the number of dots in which visual grouping task can be performed in visual noise was calculated as the median for the reversal values in which the stimulus size changed by 5 noise dots. Each task was performed three consecutive times (the final results contained three threshold values for each stimulus).

### 2.4. Procedure

The stimuli were demonstrated on TRIUMPH BOARD 89” Multi-Touch interactive whiteboard from a ceiling-affixed projector EPSON EB-685W (98 × 187 cm; 838 × 1606 px) at 70 cm. The whiteboard was attached to a grey wall to ensure that there were no other stimuli that would attract the observer’s attention. The light in the room was turned on throughout the experiment, providing photopic light conditions. The stimuli were presented in the central visual field (0 degree eccentricity) and in the peripheral visual field (15 and 30 degrees eccentric). The eccentricity of the moving objects was calculated from the geometrical center of the stimulus prior to the stimulus motion.

The participants were seated in front of the center of the screen and instructed to keep looking at the fixation cross throughout the experiment. When the object was presented in the central visual field, the fixation cross was located behind the stimulus. In case of peripheral stimuli, the objects were to the right of the fixation cross.

The stimulus size in each eccentricity was applied based on the results from the study by Ikeda et al., demonstrating that the best performance in biological motion perception in visual noise in the central visual field is obtained when the angular size of object is around 4–8 degrees [32]. Ikeda et al. also demonstrated that the best performance of biological motion perception in visual noise at 15 degrees’ eccentricity is obtained when the angular size of the object is around 16 degrees. In order to attain the best performance, slightly larger and smaller sized objects were also utilized in each eccentricity. When increasing the overall object size, the size of each dot and noise dot, as well as the distance between the dots, was adjusted proportionately. The angular size of the stimuli in each eccentricity is demonstrated in Table 1.

As demonstrated by Kalloniatis and Luu, the visual acuity at 15 degrees eccentricity is around 0.15 decimal units and around 0.07 decimal units at 30 degrees eccentricity [41]. The angular size of each element (dot) at 20 degrees eccentricity was 1.7 mm and 3.2 mm at 40 degrees eccentricity, which, based on the calculations demonstrated by Davidson [42], corresponds to an optotype detail of 0.1 degree. Since the size of each element (dot) was larger than the maximal visual acuity in each eccentricity, it was assumed that the stimuli in the visual periphery were distinguishable.

### 2.5. Data Analysis

Data analysis was conducted in IBM SPSS Statistics software. To compare performance at different visual grouping tasks and visual eccentricities, one-way ANOVA, as well as pairwise comparisons with a paired two sample *t*-test for means with a Bonferroni adjusted alpha level of 0.017 were performed.

## 3. Results

The threshold values for the number of noise dots in which different visual grouping tasks can be performed in the central visual field are represented in Figure 5. The results demonstrate that the static visual grouping task, where objects can be grouped only by similarity and proximity, can be differentiated from its scrambled version if the visual noise is 39 ± 3 dots. The dynamic visual grouping task can be performed if the visual noise is 50 ± 6 dots. The biological motion stimulus can be differentiated from its scrambled version in a higher visual noise, i.e., 94 ± 12 dots (representing the average threshold values and standard error).

The results of the dispersion analysis (one-way ANOVA) demonstrate a significant task effect on the average number of noise dots sufficient for visual grouping task performance in the central visual field (F(2, 24) = 13.5193, *p* = 0.0001). In order to compare the threshold values in pairs and determine which of the three visual grouping tasks should be performed at significantly higher or lower visual noise levels, the paired two sample *t*-test for means was performed (Bonferroni adjusted *p*-value < 0.017). The results of the *t*-test indicate that biological motion stimuli can be differentiated in a significantly higher number of visual noise dots (M = 94, SD = 37) than the static grouping task (M = 39, SD = 10; t(8) = 4.95, *p* = 0.001) and dynamic grouping task (M = 50, SD = 17; t(8) = 3.83, *p* = 0.005). However, no significant differences were observed when comparing the threshold values in the static (M = 39, SD = 10) and dynamic (M = 50, SD = 17) visual grouping tasks (t(8) = −2.144, *p* = 0.06). The results were compared between the best performance (the highest thresholds for the number of noise dots) in each grouping task.

In order to analyze the performance of different tasks of figure–ground segmentation in the central and peripheral visual field, as well as determine whether the stimulus magnification is able to compensate for the reduced performance in the visual periphery, the results for each visual grouping task were investigated separately. In the static visual grouping task, the best performance in the central visual field was observed when the stimulus size was 4 degrees. The average number of noise dots in which the object could be differentiated from its scrambled version was 39 ± 3 dots. When observing the object in the visual periphery, the best performance in the 15 degree eccentricity was observed when the object size was 20 degrees (average number of noise dots 13 ± 2) and in 30 degree eccentricity when the object size was 16 degrees (average number of noise dots 12 ± 1) (Figure 6).

As it can be observed in Figure 6, the number of noise dots in which a static visual grouping task can be differentiated from its scrambled version is considerably higher in the central visual field than in the visual periphery. To formally test this observation and compare the performance of the static visual grouping task and the processes of figure–ground segregation in the central and peripheral visual field, one-way ANOVA was performed. The results of the dispersion analysis indicate that the visual eccentricity in which the object is demonstrated is a statistically significant factor when determining the number of noise dots in which the object can be differentiated from its scrambled version (F(2, 24) = 42.8817, *p* < 0.0001). In order to compare the threshold values in central visual field and 15 and 30 degrees of visual eccentricity, a paired two sample *t*-test for means was performed (Bonferroni adjusted *p*-value < 0.017). The results of the *t*-test indicate statistically significant differences in the threshold values when comparing the performance in central visual field (M = 39, SD = 10) and the 15 degrees of visual eccentricity (M = 13, SD = 6) (t(8) = 7.3651, *p* < 0.0001), as well as significant differences in the threshold values between the performance in central visual field (M = 39, SD = 10) and 30 of visual degrees eccentricity (M = 12, SD = 4) (t(8) = 6.4898, *p* < 0.0001). This indicates that when displayed in the central visual field, the static visual grouping task can be performed at a significantly higher visual noise level compared to when the object is shown in the visual periphery (15 and 30 degrees eccentrically). No statistically significant differences in the threshold values were observed when comparing the performances at 15 degree and 30 degree visual eccentricities (t(8) = 0.6788, *p* = 0.2).

As previously demonstrated, the number of visual noise dots in which a dynamic visual grouping task can be performed was slightly higher than in the case of the static visual grouping task (Figure 7). The best performance in the central visual field was observed when the object size was 4 degrees—the average threshold value for the number of noise dots sufficient for discriminating the object from its scrambled version was 50 ± 17. In 15 and 30 degrees of visual eccentricities, the best performance was observed when the stimulus size was 16 degrees (accordingly, the average number of noise dots was 18 ± 7 and 12 ± 3).

In order to compare the threshold values for the number of noise dots in which the dynamic grouping task can be differentiated from its scrambled version at different visual eccentricities, a one-way ANOVA was performed. Similarly, as in the results obtained in the static visual grouping task, the dispersion analysis indicates that the visual eccentricity is a significant factor when performing a dynamic grouping task with visual noise (F(2, 24) = 30.2054, *p* < 0.0001). The results of the paired two sample *t*-test for means (Bonferroni adjusted *p*-value < 0.017) demonstrate a statistically significant difference between the performance in the central visual field (M = 50, SD = 17) and 15 degrees of visual eccentricity (M = 18, SD = 7) (t(8) = 5.1013, *p* = 0.0009), as well as between the performance in central visual field (M = 50, SD = 17) and 30 degrees of visual eccentricity (M = 12, SD = 3) (t(8) = 5.8594, *p* = 0.0002). Indicating that when shown in the central visual field, the dynamic visual grouping task can be performed at a significantly higher visual noise level compared to when the object is demonstrated in the visual periphery (15 and 30 degrees eccentrically). Contrary to the results obtained in the static visual grouping task, the threshold values obtained at 15 and 30 degrees of visual eccentricities also seem to be significantly different (t(8) = 2.8229, *p* = 0.011).

When comparing the performance of different grouping tasks, we observed that biological motion can be differentiated from its scrambled version at a higher level of visual noise than static and dynamic visual grouping stimuli (Figure 6). When demonstrating the biological motion stimuli in the central visual field, the best performance was obtained when the object size was 16 degrees—the average threshold for the number of noise dots sufficient for differentiating biological motion from its scrambled version was 94 ± 12 dots. In 15 and 30 degrees of visual eccentricity, the best performance was also observed when the stimulus size was 16 degrees (accordingly 24 ± 4 and 13 ± 3 noise dots) (Figure 8).

The results of the dispersion analysis (one-way ANOVA) comparing the threshold values for the number of visual noise dots in which biological motion can be differentiated from its scrambled version demonstrate a statistically significant difference in the values obtained at different visual eccentricities (F(2, 24) = 33.7233, *p* < 0.0001). In order to compare the threshold values at the central visual field and 15 and 30 degrees of visual eccentricities, the paired two sample *t*-test for means was performed (Bonferroni adjusted *p*-value < 0.017). The results of *t*-test indicate on statistically significant differences in the obtained threshold values when comparing the performance in the central visual field (M = 94, SD = 37) to the performance at 15 degrees of visual eccentricity (M = 24, SD = 12) (t(8) = 6.8374, *p* < 0.0001), as well as when comparing the performance in the central visual field (M = 94, SD = 37) and 30 degrees of visual eccentricity (M = 13, SD = 9) (t(8) = 6.0249, *p* = 0.0002). However, no statistically significant differences were observed when the performance of biological motion perception was compared with its scrambled version at 15 and 30 degrees of visual eccentricity (t(8) = 2.3489, *t* = 0.02).

## 4. Discussion

Within the interrelated processes of visual grouping and figure–ground segmentation, the visual system discriminates the available visual information into foreground objects and the background. Perceptual organization enables the binding together of several elements into the percept of a unified object. The rest of the visual elements that are not grouped together to form the figure or foreground object do not take any shape and extend behind the foreground object. Figure–ground segmentation is a core process of perceptual organization, the mechanisms of which are still being studied extensively [43,44,45]. Additionally, recent studies indicate that the performance in figure–ground segmentation can also be applied to diagnostic tasks, such as for neurological impairments, e.g., screening for posterior cortical atrophy [46] or as a part of autism assessment procedures [47,48].

Studies analyzing figure–ground segmentation have demonstrated different parameters and stimulus characteristics that determine whether the available information could be grouped together to form the figure. These parameters are stimulus size, symmetry, proximity, curvature, shape, spatial frequency, motion, etc. [1,49]. The process of visual grouping determines which elements will be grouped together, while the process of figure–ground segmentation determines the overall shape and spatial characteristics of the object. A different question is what is the dynamics of perceptual organization? As described by Wagemans et al. and Kimchi, figure–ground segmentation occurs simultaneously with the processes of visual grouping and forms the overall perception of visual characteristics of the object [1,50]. In contrast, other models assume the sequential or hierarchic stage-wise processing of the grouping followed by figure–ground segmentation [51,52].

The present study explores the processes of visual grouping and figure–ground segmentation in the central and peripheral visual field, demonstrating that static visual grouping task in the central visual field can be performed when the visual noise level is less than forty dots. When the same group of elements moves in the same direction, the number of visual noise dots in which the object can be grouped together becomes slightly higher. Although the differences between the performance in static and dynamic visual grouping tasks were not statistically significant, the slightly increased performance is consistent with the findings of other studies, demonstrating that different grouping laws can combine and increase the performance compared to stimuli that can be grouped only by one law of perceptual organization [53,54].

Even though the processes of biological motion perception and dynamic visual grouping (proxy–object perception) are subjected to both motion perception, as well as grouping by proximity and common fate, different studies have demonstrated that the processes of biological motion perception and simple motion are not as similar as they may seem. Schenk and Zihl analyzed biological motion and coherent motion perception in patients with different brain lesions affecting motion perception. Patients with bilateral parietal lobe lesions and impaired biological motion perception in visual noise had no difficulty in perceiving coherent motion and figure–ground segmentation in static stimuli [55]. By analyzing brain activity in biological motion and coherent motion tasks, Grossman et al. demonstrated that during coherent motion tasks, brain activity is concentrated in the MT (middle temporal) area, while during biological motion perception tasks, brain activity is located at a higher point in the MST (medial superior temporal) area, which is also involved in analyzing optic flow information [56], thus emphasizing the different type of information processing in both of the tasks.

Neri et al. demonstrated that the sensitivity for biological motion and translation motion detection in visual noise (determined by the maximum tolerable noise) is similar in both types of motion tasks [57]. Our study, however, applied a slightly different experimental setup, resulting in clear and substantial differences between the perception of simple motion and biological motion in visual noise: according to our results, biological motion can be perceived in a significantly larger amount of visual noise dots than simple motion. As demonstrated by Wagemans et al. and Vecera et al., previous experience with similar types of stimuli may improve performance in figure–ground segmentation tasks, which might have caused the increased sensitivity in biological motion perception compared to the simple motion task [1,58]. The differences between the results of the current study and those by Neri et al. may also be linked with different experimental setup, i.e., in the study by Neri et al., the dots constructing the biological motion object were constantly shifting, and the parameters of the applied visual noise were not specified [57].

The differences in the performance of biological motion perception might also be related to the specific perceptual processes of human motion detection in a similar manner to face recognition, which is considered to be a specific process of perceptual organization and grouping [58,59]. A recent study by Cao and Handel (2019) demonstrated that body motion, i.e., walking, enhances visual processing in the peripheral visual field. Increased contrast sensitivity for peripheral compared to central stimuli was observed when subjects were walking compared to subjects standing still [60]. Pitcher and Ungerleider even proposed that in addition to ventral and dorsal neural pathways, there is a third visual pathway that is specialized for the dynamic aspects of social perception, i.e., biological motion perception [61].

Cutting et al. were among the first to demonstrate that biological motion can also be perceived in visual noise. By using two levels of visual noise (22 and 55 dots), Cutting et al. demonstrated that observers are capable of detecting biological motion in 22 noise dots [62]. Neri et al. compared the perception of biological motion and translation motion and demonstrated that biological motion object can be perceived even when the visual noise is constructed of more than 1000 dots [57]. However, as previously mentioned, the visual noise parameters applied by Neri et al. were not entirely clear. In the study by Ikeda et al., biological motion stimuli were also embedded in visual noise, and the participants were instructed to determine which of the previously demonstrated biological motion stimuli was presented to them. The results by Ikeda et al. demonstrated that the biological motion discrimination task was able to be performed at around 55–60 noise dots [32]. Thompson et al. demonstrated a slight decrease in biological motion perception when the visual noise reached 61 dots. However, the performance in biological motion detection was still above 50% even when the visual noise reached 180 dots [34]. The results of the current study indicate that biological motion can be detected when the visual noise is constructed of around 94 dots. These results are close to Thompson et al.; however, they are not entirely equal. It seems that the observed differences in the maximum tolerable noise level between this and other previously mentioned studies might be attributed to the visual noise density that might be a slightly more informative measurement than the number of visual noise dots.

A substantial part of the current study analyzes the processes of figure–ground segmentation in the visual periphery (15 and 30 degrees). By proportionally increasing the size of each element, the whole stimulus, and the size each noise dot, the current study examines whether stimulus magnification is sufficient for equalizing the performance in central and peripheral vision. Similarly to the study performed by Ikeda et al., the chosen criterion for performance analysis was the number of visual noise dots in which the object could be differentiated from its scrambled version [32]. The results demonstrate that in all three visual grouping tasks (static, dynamic, and biological motion) the performance was significantly worse in the visual periphery than in the central visual field. Object magnification could not compensate for the reduced performance in any of the three grouping tasks, even though the objects were large enough to be perceived based on the peripheral visual acuity. These findings extend the results by Ikeda et al. demonstrating that stimulus magnification can not only compensate for the reduced performance of biological motion perception in the peripheral visual field, but also for the reduced performance of static and dynamic motion perception tasks in the peripheral visual field. Gurnsey et al. [36] indicated that the reduced performance demonstrated by Ikeda et al. [32] could be attributed to global information analysis while biological motion perception in the peripheral visual field could be based on the analysis of local cues. The results of the current study extend the analysis by Gurnsey et al. [36], indicating that the global information analysis in the peripheral visual field might be attributed to difficulties in grouping and figure–ground segmentation.

There is still no common ground regarding the relationship between the processes of figure–ground segmentation and direct attention. Treisman and Julesz (1981) have described the processes of figure–ground segmentation as pre-attentive, indicating that certain elements of the visual information emerge automatically without directing attention to them [63,64]. However, there are studies that demonstrate a clear relationship between figure–ground segmentation and direct attention, indicating that directing the gaze to the stimuli can significantly improve the performance of figure–ground segmentation. One of these studies was performed by Peterson and Gibson exploring the capability of maintaining a stable percept of an object after figure–ground segmentation, depending on the observers’ gaze location. The results demonstrated that fixating directly the object (direct attention) significantly increases the stability of the perceived object and that object recognition precedes figure–ground segmentation [65]. Klatt and Smeeton analyzed the performance of different peripheral visual tasks in young soccer players and demonstrated that the performance of object-detection tasks did not decrease towards the visual periphery. However, the performance of other attentional tasks, e.g., postural detection, decreased towards the peripheral visual field [66]. The results obtained by Peterson and Gibson [67] and Klatt and Smeeton [66], as well as in the review by Wagemans et al. [1] and the results obtained in the current study, all clearly indicate that figure–ground segmentation is a post-attentive process.

Although there are studies demonstrating that it is possible to perform a figure–ground segmentation task without direct attention [49,67], Kimchi [50] demonstrated that the role of direct attention in these tasks might also be related to different laws of visual grouping (objects can be grouped both pre-attentively as well as post-attentively). The study by Kimchi analyzed task performance while the information on the visual periphery was grouped by different visual grouping laws [50]. According to these results, the task performance in the central visual field was significantly weaker when the objects in the visual periphery were grouped by similarity and distinguished from the surrounded visual noise. The above described results of different studies together with the results of the current study indicate that the figure–ground segmentation requires direct attention and the performance can be improved when the object is demonstrated in the central visual field. However, this does not mean that the visual grouping and figure–ground segmentation does not occur in visual periphery; according to our results, these processes are significantly weaker but, first, depend on stimuli type (the best performance is in case of biological motion stimuli) and, second, can be partially compensated by increasing the elements in visual field.

## 5. Conclusions

Biological motion perception is a specific task of perceptual organization, during which the percept of a moving human body is formed based only on the information on the major joint movements [5]. The human visual system is highly sensitive to biological motion perception, and as previously demonstrated by other studies, biological motion stimuli can be detected even when the object is represented by a reduced number of dots or when the object is embedded in visual noise [11,68]. Pitcher and Ungerleider even proposed that in addition to ventral and dorsal neural pathways, there is a third visual pathway that is specialized for the dynamic aspects of social perception, i.e., biological motion perception [69]. This also fits the approach that assumes motion to be a core factor in visual segmentation [70]. The findings of the current study provide evidence that biological motion stimuli can be detected in higher levels of visual noise than static and dynamic visual grouping tasks, once again supporting the significance of human (and animal) motion signals.

The results of the current study also expand the existing knowledge on perceptual organization, particularly regarding the processes of figure–ground organization in the peripheral visual field by demonstrating that while tasks requiring figure–ground organization can be performed across different visual eccentricities, the highest accuracy can be achieved only when the objects are shown in the central visual field. However, this observation most likely exceeds biological motion perception and applies to the principles of perceptual organization in general.

While the results of the current study demonstrate a strong statistical effect, several limitations of the study should be addressed: (1) our study had a small sample size and (2) specific neurological and psychiatric conditions were not examined, except for a brief questionnaire prior to the study. The results of the current study should therefore be considered as preliminary and further studies are necessary in order to attain a more detailed picture of figure–ground segmentation mechanisms in different parts of the visual field.

## Figures and Tables

**Figure 1 brainsci-13-00380-f001:**
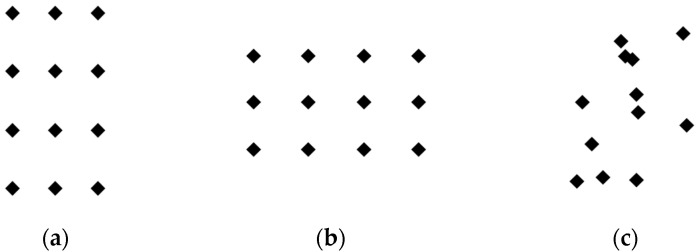
Stimuli for the static visual grouping task. The task was to determine whether (**a**,**b**) the 12 dots were distributed in columns or (**c**) the scrambled version of the stimulus was demonstrated.

**Figure 2 brainsci-13-00380-f002:**
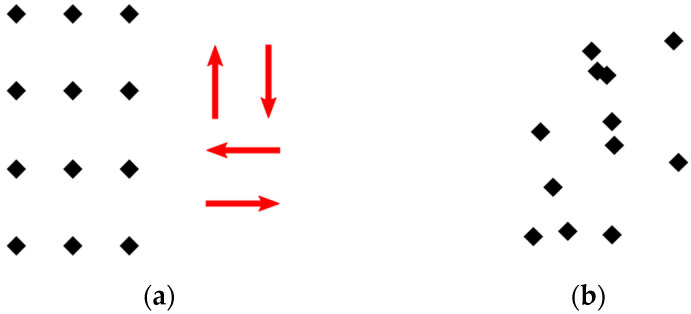
Dynamic visual grouping task. The task was to determine whether (**a**) all of the 12 dots were distributed in three columns and simultaneously moving in one of four directions (up, down, or to the left or right) or (**b**) the scrambled version of the stimulus was demonstrated, i.e., the dots were not organized in columns and all of the dots were randomly moving in different directions.

**Figure 3 brainsci-13-00380-f003:**
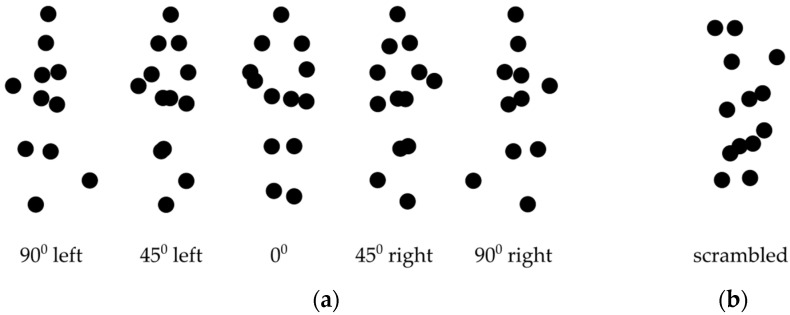
(**a**) Biological motion stimulus and (**b**) its scrambled version. Both stimuli are generated from 13 dots of the same size and moving trajectory. In the case of the scrambled version, the position of each dot is relocated so that the dots cannot be grouped as a point light walker.

**Figure 4 brainsci-13-00380-f004:**
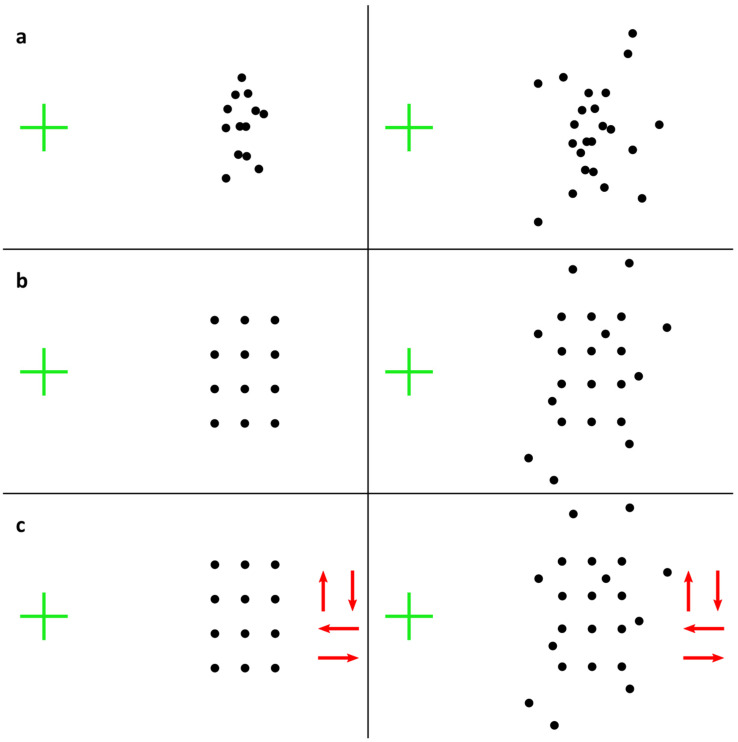
Stimuli embedded in visual noise. Participants were presented with (**a**) biological motion, (**b**) static, and (**c**) dynamic visual grouping stimuli embedded in visual noise. Each point of the visual noise had similar characteristics (size, color, and motion trajectory) as the points of the corresponding scrambled version of the stimulus.

**Figure 5 brainsci-13-00380-f005:**
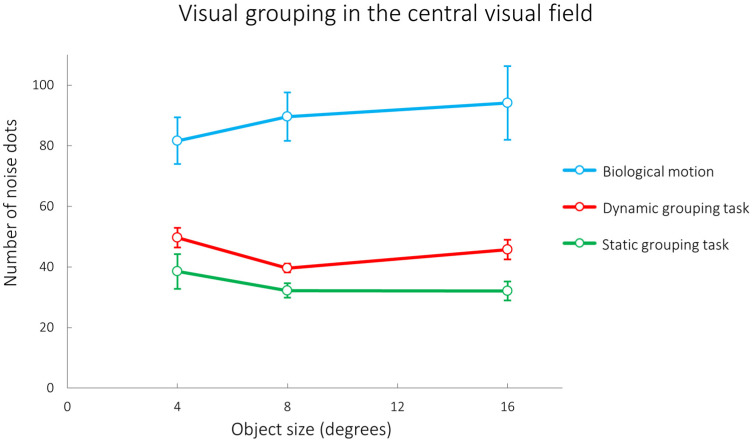
The average threshold values for the number of noise dots in which visual grouping objects can be differentiated from its scrambled version in central visual field. Data represent the average threshold values and the standard error.

**Figure 6 brainsci-13-00380-f006:**
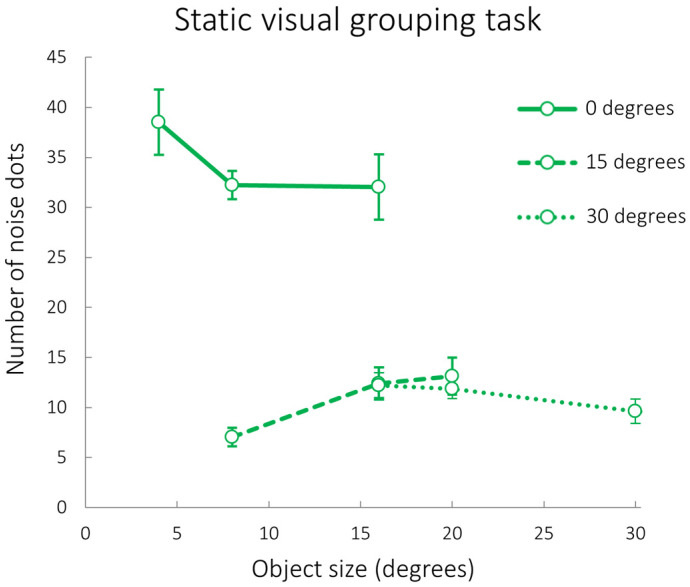
Static visual grouping task in central and peripheral visual field. Figure represents the average threshold values and standard deviation for the number of noise dots in which visual grouping stimulus can be differentiated from its scrambled version.

**Figure 7 brainsci-13-00380-f007:**
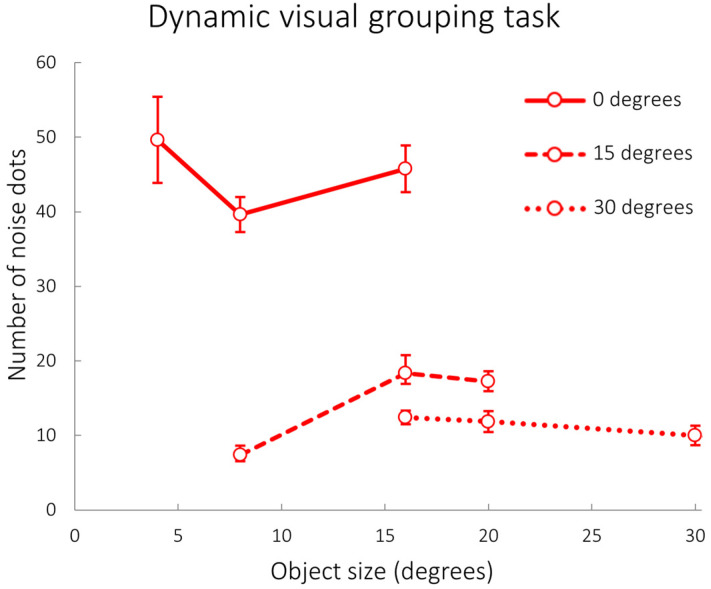
Dynamic visual grouping task in central and peripheral visual field. Figure represents the average threshold values and standard deviation for the number of noise dots in which visual grouping stimulus can be differentiated from its scrambled version.

**Figure 8 brainsci-13-00380-f008:**
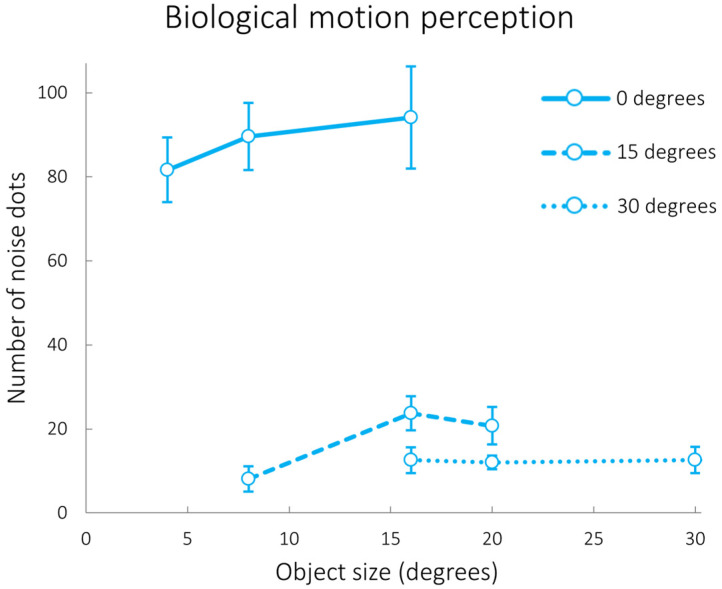
Biological motion perception in central and peripheral visual field. The figure represents the average threshold values and standard deviation for the number of noise dots in which visual biological motion stimulus can be differentiated from its scrambled version.

**Table 1 brainsci-13-00380-t001:** Stimulus size in the central and peripheral visual field.

Visual Eccentricity (Degrees)	Stimulus Size (Degrees)					
0	4	8	16			
15		8	16	20		
30			16	20	30	40

## Data Availability

The data presented in this study are available on request from the corresponding author. The data are not publicly available due to privacy reasons.

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
