# Peer review of "Figure–Ground Segmentation and Biological Motion Perception in Peripheral Visual Field"

_brainsci, 2023, doi:10.3390/brainsci13030380_

Round 1

Reviewer 1 Report

The manuscript proposes a method to explore the processes of visual grouping and figure-ground segmentation in central and peripheral visual field, expanding the understanding of perceptual organization across different eccentricities. Three tasks were designed in order to analyze figure-ground segmentation in central and peripheral visual field: (1) a static visual grouping task, (2) a dynamic visual grouping task, and (3) a biological motion detection task. The stimuli in (1) and (2) were generated from 12-13 dots grouped by proximity and common fate, and, in (3), light dots representing human motion. To analyze the processes of figure-ground segmentation the stimuli were embedded in motion noise; the threshold value for the number of dots sufficient for perceiving visual grouping was determined. This is an interesting paper. There are some suggestions for revision.

1)      At the end of the abstract, it will be more intuitive and convincing to illustrate the qualitative results of a large number of experiments for verifying the superiority and effectiveness.

2)      At the abstract, the thesis mentions that “The results demonstrate that biological motion can be distinguished from the scrambled set of moving dots in a more intensive visual noise than static and dynamic visual grouping tasks”. How is the biological motion can be distinguished?

3)      The motivation is not clear. Please specify the importance of this paper.

4)      Please highlight the contributions/innovations of this paper.

5)      Figure 2 represents dynamic visual grouping task. Please explain the formation process and specific meaning of A and B in the image.

6)      More technical details used in this paper should be given.

7)      What is the simulation environment?

8)      The simulation results are not convincing. Please add more results of comparison experiments.

9)      Make sure your conclusions appropriately reflect on the strengths and weaknesses of your work, how others in the field can benefit from it, and thoroughly discuss future work.

10)    Most of references are out of date. In the reference section, it will be better to cite more latest research, which can better reflect the novelty of this paper.

Author Response

January 18th, 2023

Response to Reviewer #1

Thank You for considering the manuscript “Figure-ground segmentation and biological motion perception in peripheral visual field” for publication at Brain Sciences. Here are our responses for the comments and recommendations of the Reviewer #1 [the updated version of the manuscript can be found in the attachment):

1) At the end of the abstract, it will be more intuitive and convincing to illustrate the qualitative results of a large number of experiments for verifying the superiority and effectiveness.
2) At the abstract, the thesis mentions that “The results demonstrate that biological motion can be distinguished from the scrambled set of moving dots in a more intensive visual noise than static and dynamic visual grouping tasks”. How is the biological motion can be distinguished?

We have revised the abstract, by focusing more on the scientific basis and the general conclusions of the current study. The word “distinguished” has been replaced with the word “differentiated” throughout the manuscript.

  1. 3)  The motivation is not clear. Please specify the importance of this paper.

  2. 4)  Please highlight the contributions/innovations of this paper.

We have tried to emphasize the motivation of the study in the Abstract, Introduction, as well as in the Conclusions part which has been added to the manuscript after the revisions. The Conclusions part includes the contributions and innovations of the current study.

5) Figure 2 represents dynamic visual grouping task. Please explain the formation process and specific meaning of A and B in the image.

The description of the Figure 2 was incomplete. The description did not include information on how the dots were moving in the scrambled version of the stimuli (B). The dynamic grouping task involved 12 dots that were distributed in three columns and that were simultaneously moving in one of four directions (up, down, to the left or right) (A). The scrambled version of the stimuli (B) consisted of the same number of dots (12), however, the dots were not organized in columns and all the dots were randomly moving in different directions.

We have made changes in the description of Figure 2

  1. 6)  More technical details used in this paper should be given.

  2. 7)  What is the simulation environment?

  3. 8)  The simulation results are not convincing. Please add more results of comparison experiments.

Additional details of the procedure and the data analysis have been added to the manuscript (sections Participants, Procedure and Data analysis).

Ilze Ceple University of Latvia Raina blvd. 19, Riga, Latvia LV-1586

9) Make sure your conclusions appropriately reflect on the strengths and weaknesses of your work, how others in the field can benefit from it, and thoroughly discuss future work.

We have added the Conclusions part to the manuscript, including an analysis reflecting the strengths and weaknesses of the manuscript.

10) Most of references are out of date. In the reference section, it will be better to cite more latest research, which can better reflect the novelty of this paper.

We have added some up-to-date references (5 years or younger) reflecting the novelty of the paper. We have also improved the language and corrected different inaccuracies throughout the paper.

Once again, thank You for Your time and effort, as well as all the suggestions that improved the value of the current manuscript!

Sincerely, Ilze Ceple

Reviewer 2 Report

The manuscript reports an interesting study about the visual perception and cognitive evaluation of movements. The paper is well written. The introduction is very clear and explains the field well. However, I have some concerns that should be considered by the authors regards their results:

- please, include in the methods a description of your statistical plan.

- your sample is very small. Have you evaluated this aspect before the study? How did you decide to include only 9 people? Why 8 females and only 1 man? These are aspects that take down enthusiasm from your results so they need a justification.

- How were enrolled participants? How much did they know about the field? Were inclusion/exclusion criteria only based on visual accuracy? What about neurological and psychiatric disorders that might compromise visual-spatial abilities (like eating disorders)? 

- line 290 something is missing

Author Response

January 18th, 2023

Response to Reviewer #2

Thank You for considering the manuscript “Figure-ground segmentation and biological motion perception in peripheral visual field” for publication at Brain Sciences. Here are our responses for the comments and recommendations of the Reviewer #2 (the newest version of the manuscript can be found in the attachment):

1. please, include in the methods a description of your statistical plan.
We have added a separate Data analysis part under the section Materials and methods, including the information on the statistical methods applied in the current study.

2. your sample is very small. Have you evaluated this aspect before the study? How did you decide to include only 9 people? Why 8 females and only 1 man? These are aspects that take down enthusiasm from your results so they need a justification.
Unfortunately, the aspect of the sample size was not evaluated before the study which is also one of the biggest limitations of the current study. Also, we did not find any previous research work on the gender differences in the processes of figure-ground segmentation. However, we assume that the results would show a similar pattern in a larger sample based on the strong statistical effect achieved in the current study.

3. How were enrolled participants? How much did they know about the field? Were inclusion/exclusion criteria only based on visual accuracy? What about neurological and psychiatric disorders that might compromise visual-spatial abilities (like eating disorders)?
The participants were bachelor students of Optometry, University of Latvia (which are mainly female students). Participants did not have any experience in with similar studies, they were not informed about the aims of the study until after the task performance. Participants also hadn’t taken a course of Vision Science yet, and they had minimal knowledge in the processes of perceptual organization. Before the task performance they were shortly asked about any neurological or psychiatric disorders that they have encountered with (without specifications) and their visual acuity was tested.

We have updated the information about participant enrolment in the manuscript.

4. line 290 something is missing
This error has been corrected. We have also improved the language and corrected different inaccuracies throughout the paper.

Once again, thank You for Your time and effort, as well as all the suggestions that improved the value of the current manuscript!

Sincerely, Ilze Ceple

Round 2

Reviewer 1 Report

There is no significant improvement in the revised manuscript. More technical details of the propsoed solution should be given, such as equations and mathematical proofs. The experimental results are still not convincing. Please compare the proposed solution with more recently published solutions. More solutions published in 2022 should be discussed. 

Reviewer 2 Report

I think the authors should moderate their enthusiasm about the results that should be considered preliminary. Nothing might be assumed about the future study because 9 participants are really a small number. I think that the paper to be considered for publication should be revised as a "preliminary study", stating clearly all the limits of the study (e.g., no evaluation for neurological or psychiatric conditions - if you did not evaluate any conditions - and unbalanced for sex). Moreover, t-tests were applied. Have the authors evaluated the parametric distribution of the data? They stated that they corrected with Bonferroni, but they did not report the p-value considered for statistical significance. Finally, in the text, there are also ANOVA analyses that are not reported in the statistical plan. 
